# Question Answering and Question Generation for Finnish

**Ilmari Kylliäinen** and **Roman Yangarber**
University of Helsinki, Finland
Department of Digital Humanities
`first.last@helsinki.fi`

## Abstract

Recent advances in the field of language modeling have improved the state-of-the-art in question answering (QA) and question generation (QG). However, the development of modern neural models, their benchmarks, and datasets for training them has mainly focused on English. Finnish, like many other languages, faces a shortage of large QA/QG model training resources, which has prevented experimenting with state-of-the-art QA/QG fine-tuning methods. We present the first neural QA and QG models that work with Finnish. To train the models, we automatically translate the SQuAD dataset and then use normalization methods to reduce the amount of problematic data created during the translation. Using the synthetic data, together with the Finnish partition of the TyDi-QA dataset, we fine-tune several transformer-based models to both QA and QG and evaluate their performance. To the best of our knowledge, the resulting dataset is the first large-scale QA/QG resource for Finnish. This paper also sets the initial benchmarks for Finnish-language QA and QG.

## 1 Introduction

The purpose of question answering (QA) systems is to help users find information more efficiently. QA systems come in many forms and offer help in everything from database querying to complex information search from the entire World Wide Web. Recently, much attention has been directed toward developing extractive QA models that can draw answers directly from spans of text. Popular approaches have emerged that integrate components that first retrieve documents relevant to a question, with models for reading comprehension that pinpoint the answers in the retrieved documents.

A task closely related to QA, yet less researched, is question generation (QG), where the object is to generate natural and grammatical questions that can be answered by a specific answer using some given context. QG can be used to, e.g., automatically create reading comprehension tasks, or to improve the interactivity of virtual assistants. It can also be used as a data augmentation tool—to create new training data for QA systems.

Recently, the focus for both tasks has moved to neural language models utilizing transfer learning—e.g., BERT (Devlin et al., 2019) or XLNet (Yang et al., 2019), at least for languages such as English. Despite the advances in QA and QG, the lack of training datasets has hindered the use of state-of-the-art deep learning methods to develop modern QA and QG models for Finnish. Finnish, like many languages, lacks the resources to train models for the two tasks. In fact, no monolingual Finnish QA or QG models have been reported to exist at all.

In order to fine-tune models for Finnish extractive QA and answer-aware QG, we first create a Finnish QA dataset by automatically translating the SQuAD—**S**tanford **Q**uestion **A**nswering **D**ataset dataset (Rajpurkar et al., 2016), from English to Finnish, and then use automatic normalization to clean up problematic data. We use the synthetic data to train several transformer-based models for QA and QG and evaluate their performance. We release the data to the research community to support future research.[1]

The paper is organized as follows: in Section (2) we review prior work on QA, QG, and generation of synthetic resources. In Section 3, we review the dataset creation, and introduce additional datasets used to train and evaluate the models. Section 4 reviews the fine-tuning methods, and Section 5 discusses the results of the experiments. Section 6 concludes and offers directions for future work.

---

[1] `https://huggingface.co/datasets/ilmariky/SQuAD_v2_fi`

## 2 Related Work

### 2.1 QA and QG for Other Languages

Approaches to both question answering and question generation have significantly evolved throughout their history. More recently, along with new datasets and novel deep learning methods, neural approaches have become the state of the art for both tasks.

It has become popular for information retrieval-based QA systems to incorporate a neural machine reading comprehension (MRC) component that extracts answers from a set of retrieved documents. After the introduction of the transformer architecture, models like BERT (Devlin et al., 2019) have become a popular tool for the answer extraction task. Many models have already surpassed human performance on the SQuAD1.1 dataset (Yamada et al., 2020; Yang et al., 2019) and some models can also predict whether the passage contains the answer to the question at all (Zhang et al., 2020). Lee et al. (2019) presented a unified end-to-end architecture capable of both retrieving and reading.

Since the mid-2010s, many RNN-based approaches have been proposed to QG (Zhou et al., 2017; Du et al., 2017; Zhao et al., 2018). However, the Transformer architecture (Vaswani et al., 2017) solved many problems that RNNs have, and has also become a popular architecture for QG models. The QG system by Wang et al. (2020) employs the encoder and the decoder from the Transformer. They combine the question generation and answer selection process in a joint model and treat the answers as a hidden pivot for question generation. Durmus et al. (2020) fine-tune a pre-trained BART model (Lewis et al., 2020) to generate questions from sentences. Chan and Fan (2019b) fine-tune a BERT model to work in a sequential manner to generate questions from paragraphs of text. Their model achieved state-of-the-art results in paragraph-level QG.

### 2.2 QA and QG for Finnish

Very little research on Finnish QA exists to date. Aunimo et al. (2004) presented two cross-lingual QA systems, `Tikka` and `Varis`, that took Finnish questions as input and found answers to them from a collection of English-language documents. `Tikka` is a simple baseline model, while `Varis` is more sophisticated. The pipelines of both systems start with defining the question type with the use of syntactic information and then translating the question into English. `Varis` also tries to extract the answer type of the question using a named entity recognizer. `Tikka` and `Varis` could correctly answer 22.5% and 29.0% of the questions presented to them, respectively.

No previous work is found on monolingual or cross-lingual QG systems that work with Finnish. Therefore, to the best of our knowledge, the results reported in this paper are the first ones for Finnish-language question generation.

### 2.3 Generation of Synthetic QA Corpora

Large annotated corpora are essential for fine-tuning pre-trained deep architecture but, unfortunately, they are also scarce for Finnish. In the context of QA, generation of synthetic corpora often means creation of a dataset via, e.g., automatic or semiautomatic translation of an existing QA dataset, or automatic data extraction from raw unlabeled data.

Recently, there have been several attempts to create synthetic datasets for QA. Carrino et al. (2020) translated an English QA dataset automatically to Spanish using a method called Translate-Align-Retrieve. The method is based on MT and an unsupervised alignment algorithm. Alberti et al. (2019) combined QG and answer extraction models with a technique they refer to as roundtrip consistency-ensuring filtering to automatically create a synthetic English QA dataset from unlabeled text passages. Abadani et al. (2021) translated the SQuAD2.0 QA dataset (Rajpurkar et al., 2018) automatically into Persian, and then finalized the data into two datasets, of which one is corrected manually and the other automatically. The automatically corrected one is many times bigger and also yielded better results. The SQuAD dataset has also been automatically translated to Swedish (Okazawa, 2021) and French (Kabbadj, 2018).

## 3 Data

### 3.1 SQuAD

SQuAD is a large English QA dataset created for training machine learning models for the extractive QA task. It is one of the most popular QA datasets, and many other QA datasets have followed its methodology (Clark et al., 2020; d'Hoffschmidt et al., 2020; Lim et al., 2019). SQuAD has also been a popular resource for answer-aware neural question generation (NQG) (Chan and Fan, 2019a; Du et al., 2017; Klein and Nabi, 2019).

|  | **English** | **Finnish translation** |
|---|---|---|
| **Passage** | The capital, Brazzaville, is located on the Congo River, in the south of the country, immediately across from Kinshasa, the capital of the Democratic Republic of the Congo. | Pääkaupunki Brazzaville sijaitsee Kongo-joen varrella maan eteläosassa, vastapäätä Kongon demokraattisen tasavallan pääkaupunkia Kinshasaa. |
| **Question** | What country does Kinshasa serve as capital of? | Minkä maan pääkaupunki Kinshasa on? |
| **Answer** | Democratic Republic of the Congo | Kongon demokraattinen tasavalta |

Table 1: An example of problematic data resulting from translating passages and answers separately. The translated answer (in the nominative case) is not found within the translated passage (where it appears in the genitive case) which is required for extractive QA.

The first version of SQuAD (SQuAD1.1) contains over 100K passage-question-answer triplets that crowdworkers extracted from 536 Wikipedia articles. Each article is divided into several passages, and each passage has several questions related to its contents. Each question is linked with an answer (a substring of the passage) and the position of the answer's first character in the passage. The second version of the dataset, SQuAD2.0, contains additional 50K questions, similar to the first version's questions but impossible to answer with the given passage. The extension's idea was to enable the development of models that can identify unanswerable questions.

### 3.2 Dataset Translation and Normalization

We translated all the text data in the SQuAD2.0 into Finnish using Google NMT (Wu et al., 2016) with the Google Translate API. The passage, questions, and answers were translated separately, which led to many of the translated answers not being substrings of the translated passage. That was sometimes caused by translation errors, but one major factor was that the data was translated from a weakly inflected, analytic language to a highly inflected, agglutinative language. In other words, the MT system has no way of knowing how to inflect the words in the translation without any context. The SQuAD format requires the answer to be a substring of the passage as it is an extractive QA dataset. The problem is illustrated in Table 1. Okazawa (2021) used a simple highlighting technique to tackle this problem when translating SQuAD2.0 into Swedish. Rather than translating the passage and the answer separately, they put special markers (`[0]`) around the answer substring before the translation and afterward simply extracted the translated answer span between the markers and then removed the markers. However, using it

would have required translating the same passages multiple times with different answers marked since passages are linked with several questions. This was not feasible solely because using Google NMT via API is not free.

After translation, we used simple normalization methods to identify the answer substring in the translated passage whenever it did not contain the separately translated answer. In total, there were four normalization steps: regular expressions, lemmatization, stemming, and using the English answer.. The script started with the first one and moved to the next one if necessary.

In the first step, a set of regular expressions was used to fix some inconsistencies (in, e.g., white spaces and punctuation) that were found to occasionally occur in the translations. In the next step, both the passage and the answer were lemmatized, and the script checked whether the now lemmatized answer was included in the lemmatized passage. If lemmatization did not lead to a match, the script moved to the next step: stemming. Stemming was done because the lemmatizer was observed to not recognize many of the passage words as they were often proper nouns. If no match was found after stemming, the last step was to check whether the English answer was included in the translated passage; if it was, it was used as the answer with the assumption that the English answer was mistakenly translated. This was often the case with, e.g., English song and movie names when they were translated with no context. If no match was found after all normalization, the question-answer pair was discarded from the final dataset.

If there was a match at any normalization step, the script proceeded to search its location in the passage. The answer search started from the English answer's relative position in the translated passage and continued to neighboring positions un-

til the answer was found. This was done to reduce the chance of choosing the starting position of a wrong occurrence, as some passages contain the answer string multiple times in different positions. After finding the answer start position, the question-answer pair was added to the final dataset.

With the normalization procedure, roughly 32K answers were modified to match the passage strings. The data consists of 101,120 passage-question-answer triplets that are valid in the sense that the answers are included in the passages. 66K of them are answerable (from SQuAD1.1), and 34K are unanswerable with the given passage (from SQuAD2.0). This means that roughly 28% of the data included in the publicly available partition of SQuAD1.1 (92K questions) had to be discarded. The amount is approximately the same when taking into account also the "unanswerable" questions of SQuAD2.0.

### 3.3 Finnish TyDi-QA Corpus

TyDi-QA—**T**ypologically **D**iverse **Q**uestion **A**nswering (Clark et al., 2020), consists of two QA datasets, covering 11 typologically diverse languages with 204K question-answer pairs. The data was collected from Wikipedia articles by human annotators. Unlike with SQuAD, the question writers formed questions without knowing the answers to them. The authors chose this strategy to reduce lexical overlapping between questions and passages, which could be exploited by machine learning systems.

One of the two datasets TyDi-QA consists of is in the SQuAD data format, which makes it ideal to combine with the SQuAD data. In total, it contains 7,635 Finnish questions. It is not much compared to SQuAD, but to the best of our knowledge, it is the only dataset that contains any Finnish data for extractive QA purposes. Consequently, we decided to include the Finnish partition of the TyDi-QA dataset in our experimental dataset.

### 3.4 The QA100-fi Corpus

Because most of the data used to train, validate, and test the models are synthetically generated, we decided to also create an additional small Finnish dataset for evaluation purposes only, QA100-fi. One option would have been to simply use the Finnish TyDi-QA data for evaluation. However, it would not have been feasible due to the possible differences with SQuAD questions caused by the TyDi-QA annotators not knowing the answers to their formed questions.

The QA100-fi dataset contains 100 questions related to Finnish Wikipedia articles. It is in the SQuAD format, and there are 10 questions for each category identified by Rajpurkar et al. (2016). We did not use any popularity-based ranking method to select the articles, like the authors of SQuAD did. Instead, we simply selected articles that appeared to be of good quality and had a length of at least three paragraphs. The dataset is tiny compared to actual QA test sets, but it still gives an impression of the models' performance on purely native text data collected by a native speaker.

### 3.5 Data Split

To train and evaluate models, we use data consisting of the answerable questions of the translated SQuAD1.1 data and the Finnish TyDi-QA data. Mimicking the methodology of Du et al. (2017), who used SQuAD data for English QG, we shuffled and split the data on article level into training, validation, and testing partitions. We call the resulting dataset SQuADTyDi-fi. The same SQuADTyDi-fi splits were used to train, validate, and evaluate both QA and QG models. We also use QA100-fi as an additional evaluation dataset. The split sizes are illustrated in Table 2.

| Dataset | Split | Q-A Pairs | Articles |
|---------|-------|-----------|----------|
| SQuADTyDi-fi | Train | 64,604 | 6,977 |
| | Dev | 4,903 | 567 |
| | Test | 4,822 | 567 |
| QA100-fi | Test | 100 | 67 |

Table 2: Dataset splits. `Q-A Pairs` refers to the number of question-answer pairs in the corresponding split, and `Articles` tells how many Wikipedia articles the split has data from.

## 4 Model Fine-tuning

We train three models for QA and four models for QG. As the base models for fine-tuning, we use the Finnish GPT-2[2] (Radford et al., 2019), FinBERT[3] (Virtanen et al., 2019), and multilingual M-BERT, (Devlin et al., 2019).

---

[2]`https://huggingface.co/Finnish-NLP/gpt2-medium-finnish`

[3]We use `bert-base-finnish-cased-v1`, the cased variant.

## 4.1 BERT Question Answering

To use BERT for extractive QA, we employ the method described in Devlin et al., 2019. BERT is fine-tuned to "highlight" the answer when given a question and a passage that contains the answer as input. In practice, the model's task is to output two types of probabilities for each input token: **1)** being the answer span start **2)** being the last token of the answer span.

The input consists of a passage and a question, separated with the [SEP] token:

$$X = (\texttt{[CLS]},\ \langle P \rangle,\ \texttt{[SEP]},\ \langle Q \rangle) \quad (1)$$

where $\langle P \rangle$ is the input passage sequence and $\langle Q \rangle$ is the question sequence.

## 4.2 BERT Question Generation

The BERT models are fine-tuned for QG using the BERT-HLSQG (Highlight Sequential Question Generation) method originally presented by Chan and Fan, 2019b. In BERT-HLSQG, the previous decoding results are considered when decoding the next token. Tokens are generated one by one using a strategy to modify BERT into generating text in an autoregressive manner. Another key idea in HLSQG is to highlight the answer in the input passage with special tokens to tackle any ambiguity caused by the answer appearing multiple times in the passage.

At inference, the input $X$ for an HLSQG model is in the following format:

$$X = (\texttt{[CLS]}, P_{HL}, \texttt{[SEP]}, \hat{Q}, \texttt{[MASK]}) \quad (2)$$

where $P_{HL}$ is the highlighted passage sequence and $\hat{Q}$ is the predicted question sequence.

At the first inference step, the highlighted passage is followed only by a [MASK] token, as the predicted question sequence $\hat{Q} = [\hat{q}_1, \hat{q}_2, ..., \hat{q}_{|\hat{Q}|}]$ is empty at the start. The passage highlighting is done by placing special [HL] tokens around the answer in the passage:

$$P_{HL} = (p_1, ..., \texttt{[HL]}, p_s, ..., p_e, \texttt{[HL]}, ..., p_{|P|}) \quad (3)$$

where $p_n$ is the $n$th passage token, $p_s$ and $p_e$ are the answer start and end tokens, and $|P|$ is the passage length.

During each step, the whole input is fed to the model, and it outputs a prediction for the [MASK] token. That prediction is considered the next token in the question sequence, and a new [MASK] token

is placed after it. The same procedure goes on with inputs updated with the newly predicted question tokens until a [SEP] token is predicted. At that point, the question is considered ready.

## 4.3 GPT-2 Question Answering

To fine-tune a GPT-2 model for QA (GPT-2-QA), we use a prompt to encourage the model to generate answers relevant to the given passage and question. The model should learn the pattern of the prompt and also the relation between the two input sections (passage and question) in the prompt.

During fine-tuning, the prompt consists of three lines. Each line starts with a word that describes the content of the line and is followed by a matching sequence. For example, the first two lines start with *Context:* and *Question:* and continue with the passage and question sequences. During training, language modeling loss is computed only on the section where the model should output the answer. The fine-tuning prompt is:

$$X =$$
$$Context{:}\langle P \rangle \quad Question{:}\langle Q \rangle \quad Answer{:}\langle A \rangle$$

where $\langle P \rangle$ is the passage sequence, $\langle Q \rangle$ is the question sequence, and $\langle A \rangle$ is the answer sequence. During inference, the answer sequence is omitted from the prompt, as the model's task is to fill it in.

## 4.4 GPT-2 Question Generation

We train two GPT-2-based QG models, GPT-2-QG and GPT-2-HLQG. The training and inference prompts of the GPT-2-QG model are the same as the GPT-2-QA, but the order of the last two rows is reversed. The QG models should learn to use the passage to generate a question that the second line's sequence answers. The training procedure is the same as with GPT-2-QA, but instead of answers, the training loss is computed on the generated questions. The two QG models differ in the prompts. GPT-2-HLQG also highlights the answer in the passage with [HL] tokens. The motivation for that is the same as with BERT-HLSQG: to reduce the possible ambiguity caused by the answer appearing multiple times in the passage.

## 4.5 Implementation

All the pre-trained models were accessed via the transformers[4] Python package by Hugging

---

[4] https://github.com/huggingface/transformers. Version 3.0.2 for BERT-HLSQG

Face (Wolf et al., 2020). The fine-tuning scripts were implemented using the same package along with PyTorch.[5]. For fine-tuning BERT-HLSQG models, we modified and used open-source code by Lin (2020).[6]

We fine-tune the models using two Nvidia Volta v100 GPUs and AdamW optimization with initial learning rate $5 \times 10^{-5}$. The batch size varied from 2 to 24, depending on the task and the model architecture. All the models were trained for six epochs, and a validation set was used to keep track of the training performance and thus select the best model for evaluation on the test sets. QA BERT models (`FinBERT-QA` and `M-BERT-QA`) had the best validation results after two epochs, whereas all the other models had the best validation performance after six epochs. More details regarding the fine-tuning are included in Appendix A.

## 5 Results

### 5.1 QA Results

The evaluation results for the QA models are in Table 3. The scores are multiplied by 100 to mimic the style of the official SQuAD leaderboard.[7] With both testing datasets, `FinBERT-QA` obtains the best results. However, the fine-tuned M-BERT model comes close, with EM scores 2-3% worse and F1 scores 2.8-4.5 points behind FinBERT-QA. The GPT-2 -based QA model achieves moderately good results also, but both EM and F1 scores are at least 20 points worse with both test sets.

| Dataset | Model | Exact Match | F1 score |
|---|---|---|---|
| SQuADTyDi-fi | FinBERT-QA | **58.0** | **69.9** |
| | M-BERT-QA | 56.0 | 67.1 |
| | GPT-2-QA | 37.2 | 46.9 |
| QA100-fi | FinBERT-QA | **67.0** | **83.7** |
| | M-BERT-QA | 64.0 | 79.2 |
| | GPT-2-QA | 43.0 | 56.0 |

Table 3: Evaluation of QA models on two test sets.

`GPT-2-QA` model obtained the worst results on both datasets. With an EM score of 37.2 and an F1

models and `4.8.1` for other models.

[5] Version `1.5.0+cu101` for BERT-HLSQG models and `1.9.0+cu111` for other models.

[6] https://github.com/chris4540/StudyMaskedLMForQG

[7] https://rajpurkar.github.io/SQuAD-explorer/

score of 46.9 on SQuADTyDi-fi data, it is apparent that fine-tuning has contributed to the model's ability to answer questions. The model outputs relatively short answers as expected, and it also seems to have quite well learned the expected answer type for each interrogative in the question. For example, the model mostly seems to answer questions starting with *kuka* ("who") with names/people and questions starting with *montako* ("how many") with numeral phrases. However, the results are still far behind the best-performing models.

When the question contains very different vocabulary than the passage (e.g., synonyms or idiomatic expressions), `GPT-2-QA` seems to perform particularly poorly. A closer look at the results shows that the `GPT-2-QA` model's outputs occasionally contain words that are slightly modified versions of the ones in the passage. This problem is unique to GPT-2 in the experiments as it is the only autoregressive model. Some other examples of such errors are shown in Table 4. However, most of the answers seem to be substrings of the input passages, as expected. `GPT-2-QA` seems to often fail to "understand" what specifically is being asked. Even when it seems to understand that the question should be answered with a date and the answer should be a substring of the passage, it often seems to pick just any date. And sometimes, it even modifies the date, as seen in Table 4.

| Predicted answer | Target answer |
|---|---|
| Kenji Vatanabe | Kenji Watanabe |
| 20. lokakuuta 2000 | 21. lokakuuta 2000 |
| Kypylän | Midnan kypärän |
| 3 vuotta | kolme vuotta |

Table 4: Examples of `GPT-2-QA` outputs that are not substrings of the input passage.

The other QA models, `FinBERT-QA` and `M-BERT-QA`, perform much better. They come in quite close to each other as `FinBERT-QA` outperforms `M-BERT-QA` by 2-3 points on SQuADTyDi-fi data with its EM and F1 scores of 58.0 and 69.9, respectively. The difference between the scores of `FinBERT-QA` and `M-BERT-QA` is slightly bigger with the QA100-fi test data, with which `FinBERT-QA` obtains an EM score of 67.0 and an F1 score of 83.7. Using only Finnish data and a lot larger amount of it in pre-training seems to have been beneficial for `FinBERT-QA`. Like

`GPT-2-QA`, also `M-BERT-QA` seems to occasionally struggle when the question is phrased very differently compared to the input passage.

As with `GPT-2-QA`, the longer the ground truth answer, the more likely the BERT-based models seem to predict it incorrectly. However, rather than choosing a completely wrong span, `FinBERT-QA` and `M-BERT-QA` often seemed only to pick too few words. This is also reflected in the bigger differences between EM and F1 scores of the other two models, compared to `GPT-2-QA`. Other than questions with longer answers, it is challenging to identify any specific question/answer types with which `FinBERT-QA` and `M-BERT-QA` have the most difficulties. Additional examples of outputs of the QA models are included in Appendix A.

The results of all QA models are better with the QA100-fi test dataset. It is possible that because the passages, questions, and answers in QA100-fi are not machine-translated, they could be closer to the Finnish language with which the models were pre-trained. Another factor might be the lengths of the passages, questions, and answers. Their average lengths are shown in Table 5. The passages and questions in the test partition of SQuADTyDi-fi are longer on average, but the answers are longer in QA100-fi. Longer passages are more challenging for the models as there are more tokens from which to choose the answer span start and end tokens. However, the test sets are so different in size that it is hard to say how much that affects the results.

|  | Passage | Question | Answer |
|---|---|---|---|
| SQuADTyDi-fi (test) | 74.5 | 6.6 | 2.5 |
| QA100-fi | 62.2 | 5.9 | 3.2 |

Table 5: Average word counts in the test partition of SQuADTyDi-fi and QA100-fi.

As there are no other Finnish QA models to compare with, we can gain some perspective by comparing the results with English models trained on a similar dataset. The top EM and F1 scores for single BERT models in the English SQuAD1.1 leaderboard[8] are around 85 and 90, respectively. The overall best single model results are from other transformer-based models, like LUKE (Yamada et al., 2020) and XLNet (Yang et al., 2019), which both obtain EM and F1 scores over 90

[8]Webpage mirroring SQuAD1.1 leaderboard: `https://paperswithcode.com/sota/question-answering-on-squad11`

and 95, respectively. The best Finnish results (by `FinBERT-QA`) are quite far from the best-performing English models. However, it is worth noting that the Finnish models were fine-tuned using a smaller dataset which is probably of poorer quality, as it has been automatically translated. Finnish being a highly inflective language might also make the QA task generally more challenging.

## 5.2 QG Results

The evaluation results for the QG models are in Table 6. The FinBERT-based models obtain the best results. As in the QA task, the results of the FinBERT and M-BERT-based models are quite close to each other, whereas the GPT-2 models are much worse.

| Dataset | Model | BLEU-4 | METEOR |
|---|---|---|---|
| SQuADTyDi-fi | FinBERT-HLSQG | **0.11** | **0.17** |
|  | M-BERT-HLSQG | 0.10 | 0.16 |
|  | GPT-2-QG | 0.04 | 0.10 |
|  | GPT-2-HLQG | 0.04 | 0.10 |
| QA100-fi | FinBERT-HLSQG | **0.18** | **0.22** |
|  | M-BERT-HLSQG | 0.13 | 0.20 |
|  | GPT-2-QG | 0.04 | 0.13 |
|  | GPT-2-HLQG | 0.04 | 0.11 |

Table 6: BLEU-4 and METEOR scores of QG models. Results on additional metrics in Appendix A.

Both `GPT-2-QG` and `GPT-2-HLQG` achieve a BLEU-4 score of 0.04 on both datasets. Unlike in Chan and Fan (2019b), using an answer highlight technique in the passage did not lead to an increase in the performance as the results of the two models are nearly identical. This indicates that ambiguity was not the root cause of the inferior performance of the models.

Looking at the outputs of the GPT-2-based QG models, it is clear that the models learn the general structure of a question. The outputs mostly start with the correct interrogative word and end with a question mark. The questions also seem mostly grammatical. The biggest problems seem to be related to semantic validity and generating questions that can be answered using the input answer. However, the models occasionally seem to generate questions that can be answered with the input answer, but they are very different from the ground-truth questions. They are good examples

of why using automatic, n-gram-based evaluation metrics to assess QG systems can be problematic.

Compared to the GPT-2-based QG models, the BERT-based QG models perform roughly twice as well on every metric. `FinBERT-HLSQG` and `M-BERT-HLSQG` seem to output questions that make more sense and have more common words with the target question. For example, with target question *Kuinka korkeaksi puu yleensä kasvaa avoimilla alueilla?* ("How tall does the tree usually grow in open areas?"), `FinBERT-HLSQG` outputs *Minkä korkuinen on jousisoihtupuu avoimilla alueilla?* ("How tall is the pink trumpet tree in open areas?") and `GPT-2-HLQG` outputs *Minkä kokoisia puutalot ovat metsäalueiden korkeilta tasoilta?* ("What size are the wooden houses from the high levels of the forest areas?"). `GPT-2-HLQG`'s output is nonsensical yet grammatical, whereas `FinBERT-HLSQG`'s output can be considered correct, though the phrasing is quite different from the target question. All models perform better with shorter passages and struggle at inflecting rare words. Additional examples of the outputs of all QG models are shown in Appendix A.

As on the QA task, the FinBERT-based model achieves slightly better scores on the SQuADTyDi-fi test set than the multilingual variant. However, in QG, the difference between the performance of BERT-based models is bigger when evaluating on the QA100-fi dataset. For example, `FinBERT-HLSQG` obtains a BLEU-4 score of 0.18, while `M-BERT-HLSQG` yields 0.13. Checking the outputs on QA100-fi, it seems that `M-BERT-HLSQG` has more problems inflecting words, and it occasionally uses word order and phrasings that sound a bit unnatural in Finnish. It is possible that these problems were exacerbated when the model was tested on QA100-fi, which consists of data collected by a native speaker.

Chan and Fan (2019b), who initially presented the BERT-HLSQG method, report a BLEU-4 score of 0.20 for their English QG model that was fine-tuned on roughly 73K question-answer pairs. `FinBERT-HLSQG`'s BLEU-4 score (0.11) on the SQuADTyDi-fi test set is quite far from that, whereas the BLEU-4 score on the smaller QA100-fi test set (0.18) is a lot closer. It is likely that the passages and questions in QA100-fi being shorter on average has a positive effect on the model's performance on the dataset. Chan and Fan (2019a) also conclude that their BERT-HLSQG model works better with shorter passages. As with the QA task, it is possible that the smaller amount of training data and its poorer quality, together with the more complex Finnish morphology, partly explain the differences that occur when compared to the English models.

## 6 Conclusion and Future Work

We have proposed an MT-based method for creating a Finnish QA dataset, and used it to train and evaluate several transformer-based QA and QG models. On both tasks, fine-tuned monolingual BERT models obtain the best results. The multilingual variants came close, while the fine-tuned GPT-2 models were found to underperform. Pre-training with only Finnish data seems to give the models an edge in both QA and QG.

To the best of our knowledge, these are the first monolingual Finnish QA and QG models. They set a fair baseline for further research in Finnish QA and QG. All data used in the experiments is released to the research community, to support future research, and the models are released as benchmarks. We believe that this is a valuable contribution, since suitable datasets created by native Finnish speakers are not yet available.

Given the promising initial results, we plan to pursue several directions. (1) As the SQuAD2.0 data with the unanswerable questions was also translated, it could be used to train the first Finnish QA models that can also identify unanswerable questions. (2) Lower-level natural language processing (NLP) components can be employed to study and improve performance. For example, we can use syntactic parsing to check for ungrammatical questions, to analyze the created synthetic dataset; we can use name recognition to improve QA results (Yadav and Bethard, 2019; Piskorski et al., 2019), etc. (3) Real-world applications, such as language learning systems, e.g., (Katinskaia et al., 2018, 2017), can benefit from QA and QG—by automatically generating reading comprehension questions from arbitrary authentic text. To integrate QG into such applications, a separate model should be developed for choosing the appropriate input answers. (4) To support (3), it is important to study in detail on what types questions and answers the QA and QG models do especially well or especially poorly.

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

# A Appendix

| Model | Epochs (best model) | Batch size |
|---|---|---|
| FinBERT-QA | 2 | 16 |
| M-BERT-QA | 2 | 16 |
| GPT-2-QA | 6 | 2 |
| FinBERT-HLSQG | 6 | 24 |
| M-BERT-HLSQG | 6 | 16 |
| GPT-2-QG | 6 | 2 |
| GPT-2-HLQG | 6 | 2 |

Table 7: Training hyperparameters. With all models, we use the AdamW optimization algorithm with an initial learning rate of $5 \times 10^{-5}$.

| Dataset | Model | BLEU-1 | BLEU-2 | BLEU-3 | BLEU-4 | METEOR | ROUGE-L |
|---|---|---|---|---|---|---|---|
| SQuADTyDi-fi | FinBERT-HLSQG | **0.29** | **0.21** | **0.15** | **0.11** | **0.17** | **0.33** |
| | M-BERT-HLSQG | 0.29 | 0.20 | 0.14 | 0.10 | 0.16 | 0.31 |
| | GPT-2-QG | 0.18 | 0.11 | 0.06 | 0.04 | 0.10 | 0.20 |
| | GPT-2-HLQG | 0.18 | 0.10 | 0.06 | 0.04 | 0.10 | 0.20 |
| QA100-fi | FinBERT-HLSQG | **0.39** | **0.30** | **0.22** | **0.18** | **0.22** | **0.41** |
| | M-BERT-HLSQG | 0.36 | 0.25 | 0.18 | 0.13 | 0.20 | 0.36 |
| | GPT-2-QG | 0.22 | 0.12 | 0.07 | 0.04 | 0.13 | 0.22 |
| | GPT-2-HLQG | 0.19 | 0.11 | 0.07 | 0.04 | 0.11 | 0.20 |

Table 8: All evaluation results of the QG models.

| | |
|---|---|
| **Input passage** | Ulkomuodoltaan hylkeet ovat sileitä ja pulleita. Ruumiinrakenne soveltuu sulavaan vedessä liikkumiseen. Ranteesta ja kämmenestä ovat muodostuneet etuevät ja nilkasta ja jalkaterästä takaevät. Evät ovat heikot eikä niitä voi käyttää apuna maalla liikkumiseen . Hylkeet liikkuvatkin maalla siten, että ne siirtävät painoa rinnan ja vatsan varaan. Erotuksena lähisukulaisistaan korvahylkeistä, joihin kuuluvat muun muassa merileijonat, varsinaisilla hylkeillä ei ole ulkoisia korvalehtiä. Varsinaisten hylkeiden uiminen tapahtuu evien ja ruumiin takaosan sivuttaissuuntaista liikettä apuna käyttäen. |
| **Input question** | Mihin hylkeiden evät eivät sovellu? (*What are seal fins not suitable for?*) |
| **Target answer** | maalla liikkumiseen (*to move on land*) |

| Model | Predicted Answer |
|---|---|
| FinBERT-QA | maalla liikkumiseen. (*to move on land.*) |
| M-BERT-QA | vedessä (*in the water*) |
| GPT-2-QA | ui maalla (*swim/swims on land*) |

Table 9: Output examples of the QA models. The ground truth answer is highlighted in the input passage.

| Input passage | Jättiläismetsäkarju eli jättiläismetsäsika eli jättisika (Hylochoerus meinertzha-geni) on keskisen ja läntisen Afrikan metsissä elävä elinvoimainen sorkkaeläin-laji. Se on sukunsa Hylochoerus ainoa laji. Jättiläismetsäkarjut ovat suurimpia luonnonvaraisia sikoja. Ne voivat kasvaa jopa 210 senttimetriä pitkiksi ja painaa 275 kilogrammaa. Niiden ruumis on tanakka ja pää leveä, mutta jalat ovat lyhyet. Nahkaa peittävät pitkät ja karkeat karvat, jotka nousevat pystyyn eläimen kiihtyessä. |
|---|---|
| **Input answer** | 210 senttimetriä |
| | *(210 centimeters)* |
| **Target question** | Kuinka pitkiksi jättiläismetsäkarjut voivat kasvaa? |
| | *(How long can giant forest hogs grow?)* |

| Model | Generated question |
|---|---|
| `FinBERT-HLSQG` | Kuinka pitkäksi jättiläismetsäkarju voi kasvaa? |
| | *(How long can a giant forest hog grow?)* |
| `M-BERT-HLSQG` | Kuinka pitkiä jättiläismetsäkarjat voivat kasvaa? * |
| | *(How long can giant forest cattles grow?)* |
| `GPT-2-QG` | Miten pitkäksi afrikkalainen jättiläismetsäkarju voi kasvaa? |
| | *(How long can an African giant forest hog grow?)* |
| `GPT-2-HLQG` | Kuinka pitkä on jättiläismetsäkarjun pituus? |
| | *(How long is the length of a giant forest hog?)* |

Table 10: Output examples from the QG models. The input answer is highlighted in the input passage. Outputs marked with * contain inflection errors, but they are ignored in the translation.