# OpenReview forum: "Question Answering and Question Generation for Finnish"
_NoDaLiDa/2023/Conference — NoDaLiDa 2023_

### Official Review · Reviewer_NKGS · 2023-03-08
**An interesting paper contributing a new MT'd dataset on QA for Finnish**

**Rating:** 8
**Confidence:** 4

**Review:**

This is an interesting paper, which contributes a machine-translated version of SQuAD to Finnish. The dataset is openly distributed and used to set baseline for the QA task for Finnish. The paper also introduces a small native test set. While the data is naturally noisy (thanks to the MT approach), the dataset is "much better than nothing" no doubt for the Finnish language. As machine translation quality improves, the approach of machine translating data to languages where manual creation from scratch would not be feasible is gaining in popularity and this paper contributes to our understanding of the limitations of such approach. In addition to the dataset work, the paper also introduces models both for question answering and question generation.

Minor points and questions:
1) What was the cost of translation using the Google API in this case?
2) For replicability, what were the tools used for stemming and lemmatization?
3) What is the estimate of work/time spent on language-dependent filtering, and how difficult would it be to apply the same methodology to other languages?




**Paper Type:**

Long paper

---

### Official Review · Reviewer_jbdR · 2023-03-09
**Describing the first large-scale QA/QG dataset for Finnish and presenting initial benchmarks for QA and QG in Finnish.**

**Rating:** 7
**Confidence:** 3

**Review:**

The paper describes the creating of a Finnish data set for extractive QA by translating the well known SQuAD dataset and joining it with the (much smaller) Finnish partition of the TyDi-QA corpus. Additionally they create a small test set from scratch to obtain a native set alongside the machine translated SQuAD. Several DL methods are then explored for training and testing both QA and QG on the datasets. Results are behind SOTA for English, as remarked by the authors, but present a documented benchmark for future work. The paper describes the ground work needed for lower resourced languages and contributes to the community by making the data available.
On page 7 "The results of all the QG models are better with" (line 665) probably contains a typo, as results for QA are being discussed in that section (5.1).

**Paper Type:**

Long paper

---

### Official Review · Reviewer_swBv · 2023-03-13
**Question Answering and Question Generation for Finnish**

**Rating:** 8
**Confidence:** 3

**Review:**

This paper presents (mostly automatically translated) Finnish question answering and question generation datasets and experiments to finetune Finnish LM-s to the task. Creating datasets and benchmarks for new languages is very important and the most significant contribution of this work. In general, I found the work well structured, interesting, informative and easy to read. The information was presented systematically and the authors set a new benchmark for future work.

While I'm not an expert in this particular task, I wonder if the authors should have also considered translating the test set instead and using English models given their superior performance. The also authors used Google’s translation API to create their dataset. Unfortunately, they mentioned some compromises in their methodology due to the cost of the translation service, which is surprising considering that there are also many open-source NMT models available for Finnish. Would better results be achieved if the API cost was not a factor?



**Paper Type:**

Long paper

---

### Decision · Program_Chairs · 2023-03-17

Accept